# Acute Effect of Furosemide on Left Atrium Size in Cats with Acute Left-Sided Congestive Heart Failure

**DOI:** 10.3390/ani15223267

**Published:** 2025-11-11

**Authors:** Sarah Miliaux, Alma H. Hulsman, Sanne Hugen, Niels Groesser, Erik Teske, Viktor Szatmári

**Affiliations:** 1Department of Clinical Sciences, Faculty of Veterinary Medicine, Utrecht University, Yalelaan 108, 3584 CM Utrecht, The Netherlands; a.h.hulsman@uu.nl (A.H.H.); s.hugen@uu.nl (S.H.); e.teske@uu.nl (E.T.); v.szatmari@uu.nl (V.S.); 2AniCura Specialistisch Verwijscentrum Haaglanden, Frijdastraat 20a, 2288 EZ Rijswijk, The Netherlands; niels.groesser@anicura.nl

**Keywords:** cardiomyopathy, diuretic, echocardiography, feline, point-of-care ultrasound, respiratory distress

## Abstract

Cats with heart failure often struggle to breathe because fluid builds up in or around their lungs. Because breathing difficulties can arise from several conditions, it is essential to find out the underlying cause. For this, veterinarians often use ultrasound to measure the size of the left atrium, a chamber of the heart, to help diagnose heart failure and choose appropriate treatment. Furosemide, a commonly used diuretic medication, is given to remove excess fluid from the body and with that ease breathing, but its direct effect on heart size in cats has not been studied yet. Some dyspneic cats are referred after receiving high doses of furosemide by the referring veterinarian without prior left atrium size measurements. If furosemide significantly reduced left atrium size, it could possibly lead to misdiagnosis and wrong treatment. In this study, we examined 25 cats with sudden breathing difficulties caused by heart failure. We measured the size of the left atrium and the breathing rate before diuretic treatment, and repeated this three hours after furosemide administration. We found that furosemide quickly reduced both the breathing rate and left atrial size. This study shows that while furosemide provides rapid relief, echocardiographic evidence of heart failure may be absent or underestimated, if the cat had already received diuretics before referral. Our results highlight the importance of careful history taking, especially in cats that may already have been treated with diuretics before referral.

## 1. Introduction

In veterinary emergency settings, around 25% of cats presenting with acute respiratory distress are diagnosed with acute left-sided congestive heart failure (CHF) [1]. Echocardiographic measurement of left atrial size by assessing the left atrium to aortic ratio (LA:Ao) and maximal left atrial diameter (LAD) is a key diagnostic tool in identifying CHF in cats, and differentiating CHF from primary respiratory conditions, such as feline asthma, diaphragmatic hernia, lung contusion, pneumothorax, pyothorax or pneumonia, which might have similar clinical manifestations [2,3,4]. Chronic increase in left atrial pressure leads to left atrial dilation. Cats with left-sided CHF typically have enlarged left atria because the most common underlying heart disease, hypertrophic cardiomyopathy, is a chronic condition. Increased left atrial pressure in cats can lead to either cardiogenic pulmonary edema or pleural effusion as a result of elevated venous hydrostatic pressure within the lungs [5,6].

Furosemide, a potent and fast-acting loop diuretic, is the first line treatment for pulmonary edema [7,8]. This diuretic effect decreases circulating plasma volume, leading to a reduction in hydrostatic pressure within pulmonary capillaries. Though furosemide is effective in managing pulmonary edema, in cases of large-volume pleural effusion, furosemide alone provides limited immediate clinical benefit without concurrent therapeutic thoracocentesis [7,8]. To our knowledge, no study has investigated the effect of furosemide administration on left atrial size measured by two-dimensional (2D) echocardiography in cats with left-sided CHF. Some dyspneic cats are referred after having received high doses of furosemide by the referring veterinarian without prior diagnostic point-of-care ultrasound (POCUS). This can make the diagnosis of CHF challenging due to the lack of data on the impact of furosemide on left atrial size. If furosemide significantly reduces left atrial dimensions, it could influence clinical decision-making and potentially lead to misdiagnosis, by erroneously categorizing these cats as not having CHF.

This study aims to assess whether parenteral furosemide treatment reduces left atrial size to such a degree that it would affect diagnostic accuracy and clinical management in cats with CHF. The objective was to document changes in left atrial size by evaluating the LA:Ao ratio and LAD in cats with CHF before and after furosemide administration using POCUS. We hypothesized that left atrial size would decrease after furosemide treatment, corresponding to clinical improvement of respiratory distress.

## 2. Materials and Methods

### 2.1. Animals and Inclusion Criteria

This prospective, observational, multicenter study was conducted between May 2024 and May 2025. The study was approved by the Institutional Ethics Committee (number 10813-2024-01). All client-owned cats older than one year of age that were presented with increased respiratory effort to the Small Animal Emergency and Critical Care services of the Faculty of Veterinary Medicine at Utrecht University or referral center AniCura Specialistisch Verwijscentrum Haaglanden, and were diagnosed with CHF, were prospectively enrolled.

Left-sided CHF was diagnosed based on a history of sudden onset of increased respiratory rate or increased respiratory effort, as well as an abnormal point-of-care N-terminal pro-brain natriuretic peptide (NT-proBNP IDEXX Laboratories, Westbrook, ME, USA) blood test result and abnormalities on the lung ultrasound (such as B-lines) and cardiac POCUS that were compatible with the predefined criteria for CHF (see Section 2.2) [9,10,11,12,13,14,15,16,17,18,19,20,21,22]. NT-proBNP served as a point-of-care inclusion tool to support the clinical suspicion of CHF; however, as the test was used qualitatively and not quantitatively, it was not analyzed statistically in this study. Cats were not included when (1) they received diuretics within a week prior to presentation to the emergency service; (2) had a normal or only mildly dilated left atrium size (LA:Ao < 1.8 and LAD < 18 mm); (3) the cat’s clinical condition did not allow safe further examination; or (4) the owner opted for immediate euthanasia.

### 2.2. Point-of-Care Ultrasound Examination and Study Design

The POCUS was carried out by the attending emergency veterinarians with previous experience using this modality. Participating POCUS practitioners underwent a two-hour POCUS training prior to the study, during which image acquisition competency was verified by a European Board of Veterinary Specialisation (EBVS^®^)-recognized Cardiology specialist and a resident in Emergency and Critical Care (ECC). All examinations were performed using a Philips CX50 (Philips Healthcare, Andover, MA, USA) or Esaote MyLab 7 (Esaote S.p.A., Genoa, Italy) ultrasound system equipped with a 5–10 MHz convex probe and an average frame rate of approximately 60 frames per second, ensuring adequate temporal resolution for feline cardiac imaging.

Upon presentation, cats were first stabilized in an oxygen cage (flow rate 3 L/min), resulting in an estimated inspired oxygen concentration of 40–50%. The cats were subsequently sedated using an intramuscular injection of butorphanol (Dolorex; Intervet International B.V., Boxmeer, The Netherlands) at a dose of 0.2 mg/kg, and in case of pleural effusion a thoracocentesis was performed using a 20 gauge over-the-needle catheter (Vasofix, B. Braun, Melsungen, Germany) or butterfly needle (Venofix, B. Braun, Melsungen, Germany) connected to a three-way stopcock and a 20 mL syringe. The stabilized cats were positioned either in sternal recumbency or in a standing position, and given supplemental oxygen through flow-by for the duration of the POCUS exam [22]. To minimize stress and expedite the process, cats were not shaved for the POCUS. Alcohol was applied to the fur to optimize imaging. Two-dimensional grey scale video cineloops were obtained and recorded from the right parasternal position in the standard long-axis 4-chamber and the short-axis view at the level of the heart base [22,23]. The examination and recording of cineloops was repeated three hours (T3) following parenteral furosemide (Dimazon; Intervet International B.V., Boxmeer, The Netherlands) administration by the same examiner. All left atrial measurements were reviewed from the recorded cineloops by an ECC resident, who was not present during the POCUS. Criteria for left atrium enlargement were set at a LA:Ao ratio exceeding 1.8 and a LAD greater than 18 mm (Figure 1A) [24,25,26]. Both criteria (LA:Ao ≥ 1.8 and LAD ≥ 18 mm) had to be present for inclusion; cats not meeting both criteria were excluded. Video cineloops were recorded at maximal left atrial size, and from these recordings three separate measurements were obtained; the average of these three measurements was used for data analysis.

In addition, the presence and number of B-lines per acoustic window, and the presence of pleural effusion was noted. The lung ultrasound examination was performed at four acoustic windows on each side of the cats’ thorax [16,17,18,19,20,21,22].

Following confirmation of an enlarged left atrium, cats received an initial intramuscular dose of furosemide (Dimazon; Intervet International B.V., Boxmeer, The Netherlands) at a dose of 2 mg/kg at the time of presentation (T0), administered into the epaxial muscles using a 23 G needle. Once the cats were considered clinically stable enough, an intravenous catheter was placed. Their respiratory rate and effort were monitored hourly. Additional intravenous furosemide doses (1 mg/kg) were administered hourly (T1, T2, T3) as long as the respiratory rate remained above 40 breaths per minute or increased respiratory effort persisted in the oxygen cage, within the 3 h study period. The chosen threshold of <40 breaths/min was based on published clinic-based respiratory rate data [13,14], recognizing that these measurements were obtained while cats were receiving supplemental oxygen and following mild sedation. Both interventions are known to reduce stress and respiratory effort compared to unsedated, at-home conditions. Therefore, this threshold represents a clinically stabilized in-hospital cutoff, distinct from resting home respiratory rate reference values.

### 2.3. Statistical Analysis

A priori sample size estimation was performed to determine the minimum number of cats required to detect a reduction in left atrium following furosemide administration. As no published studies have previously quantified the expected magnitude of left atrial size reduction after diuretic therapy in cats, the expected decrease was estimated by the authors based on clinical experience and pilot observations. A mean decrease of 1.5 mm with an assumed standard deviation of 1.8 mm, α = 0.05, and power of 80% indicated a required sample size of 22 cats.

Continuous variables are presented as mean ± standard deviation (SD) when the data were normally distributed, or as median and range when the data were not normally distributed, and categorical variables are expressed as counts and percentages. Normality was assessed using the one-sample Kolmogorov–Smirnov test. Variables that were not normally distributed were analysed using non-parametric tests. Paired comparisons between baseline (T0) and follow-up (T3) values were performed using the Wilcoxon signed-rank test for non-normally distributed data. Group comparisons for categorical variables were conducted using the chi-square test or Fisher’s exact test, as appropriate. To evaluate differences in LAD reduction across groups with different cumulative furosemide dosages, a simple linear regression model was used.

Statistical significance was defined as *p* < 0.05. All analyses were performed using a commercially available statistical software (SPSS, Version 30.0.0, IBM Corp., Armonk, NY, USA).

## 3. Results

### 3.1. Study Sample, Clinical and Lung Ultrasound Findings

A total of 25 cats met the inclusion criteria. Eighteen cats were neutered males, and 7 were spayed females. Domestic shorthair cats were most represented (n = 18). Other cat breeds included were British shorthair (n = 2), Sphynx (n = 2), Maine Coon (n = 1), Persian (n = 1) and Burmese (n = 1). The median age at presentation was 9 years (range 4 to 19 years). Five of the 25 cats had normal respiratory rates (<40 breaths/min) but showed a marked increase in respiratory effort [11,12,13,14,15,16]. Of these, three had cardiogenic pulmonary edema and two had pleural effusions. A heart murmur was heard in 7 cats and in only 1 cat a gallop sound was detected. The remaining 17 cats had no abnormalities on cardiac auscultation, such as murmur, gallop or arrhythmia. All cats exhibited more than three B-lines per lung field on lung ultrasound, 11 cats also showed concurrent pleural effusion. Therapeutic thoracocentesis was performed in all these 11 cats prior to the T3 measurements.

### 3.2. Correlation Between Furosemide Dose, Breathing Improvement and Left Atrial Size

Significant improvements were observed in both POCUS and clinical respiratory (rate and effort) parameters following treatment with furosemide (T3 vs. T0). The LA:Ao showed a statistically significant decrease from a mean ± SD of 2.48 ± 0.35 at T0 to 2.17 ± 0.40 at T3 (*p* < 0.001) (Figure 2A). Similarly, the LAD decreased significantly from 21.0 ± 2.8 mm to 18.4 ± 3.2 mm (*p* < 0.001) (Figure 2B). The respiratory rate improved significantly, decreasing from a mean of 64 ± 29.5 to 40 ± 14 breaths per minute (*p* < 0.001) (Figure 3). These findings suggest a favorable hemodynamic response to furosemide treatment.

Normalization of respiratory rate, defined as respiratory rate less than 40 breaths per minute at follow-up (T3), was achieved in 10 of 25 cats (50%) that were tachypneic at presentation. Normalization of LAD, defined as a measurement <18 mm, occurred in 8 of 25 cats (32%). In contrast, LA:Ao normalization (<1.8) was observed in only 1 cat, precluding further statistical analysis. To explore the relationship between cardiac decongestion and respiratory relief, a Pearson correlation was performed between the change in LAD and change in respiratory rate. No significant correlation was observed (r = −0.002, *p* = 0.991), suggesting that the magnitude of structural cardiac improvement did not parallel the degree of respiratory rate reduction in individual cats. Cumulative furosemide doses administered during the 3 h of the study varied between cats: 3 mg/kg (n = 3), 4 mg/kg (n = 2), and 5 mg/kg (n = 20). The degree of LAD reduction (T3 − T0 = Δ LAD) was compared among these groups with a regression analysis. No significant difference was detected (R = 0.108 and *p* = 0.609), indicating that Δ LAD did not correlate with the cumulative dose of furosemide.

## 4. Discussion

Our study shows that parenteral administration of furosemide in cats with acute CHF was associated with significant short-term reductions in respiratory rate and effort, as well as reduction in both LA:Ao and LAD. While LA:Ao and LAD are valuable echocardiographic indices for assessing left atrial size, they should be interpreted in combination with the overall hemodynamic status, physical examination, and other diagnostic findings, as discordance between atrial dimensions and the presence of CHF can occur. Our findings confirm the clinical efficacy of furosemide in achieving rapid hemodynamic and symptomatic improvement in feline CHF and align with previous studies where dehydration and diuresis led to decreased left atrial dimensions in cats, demonstrating the influence of preload on atrial measurements [27,28]. In particular, Campbell and Kittleson [27] demonstrated that hydration status markedly affects echocardiographic indices, including left atrial size, even in clinically normal cats. Previous studies emphasize that echocardiographic left atrial measurements are preload-dependent, and must be interpreted in the context of the patient’s volume status, especially when assessing treatment response in acute left CHF [27,28]. Importantly, this underscores the need for clinicians to consider both therapeutic interventions and hydration status when interpreting echocardiographic changes, ensuring that reductions in left atrial size are not misattributed solely to disease but also to the acute effects of diuresis [27,28].

Importantly, the finding that one third of cats in this cohort exhibited normalization of LAD within only three hours of parenteral furosemide administration indicates that left atrial size has to be evaluated cautiously in cats that received furosemide from the referring veterinarian before referral. As a result, CHF might be considered a less likely cause of respiratory distress in these cats if imaging is performed after furosemide therapy. This may lead to diagnostic challenges, particularly in referred cats that received furosemide treatment prior to initial cardiac evaluation.

Thoracic radiography remains an invaluable adjunct in the assessment of cats with suspected CHF, as it allows differentiation between cardiogenic pulmonary edema and primary respiratory causes of dyspnea. However, in this study, thoracic radiographs were not consistently performed within the defined study window because several cats were too unstable to undergo positioning and restraint safely. Point-of-care ultrasound was therefore selected as the primary imaging modality, given its feasibility, minimal stress, and ability to be performed during oxygen supplementation. Nonetheless, future studies incorporating both thoracic radiography and POCUS could provide a more comprehensive diagnostic framework for evaluating cats with respiratory distress.

Interestingly, one cat exhibited an increase in left atrium size after furosemide treatment. Possible explanations include measurement variability or an inadequate therapeutic response within the short observation window. Another possible explanation for the paradoxical increase in left atrium size includes hemodynamic or renal factors that may have limited the expected response to furosemide. Decreased renal perfusion secondary to low cardiac output or right-sided congestion could have reduced diuretic efficacy. These mechanisms could result in persistent volume overload despite furosemide administration, emphasizing that individual variation in drug response and underlying cardiorenal interactions can markedly influence short-term echocardiographic outcomes. This emphasizes that echocardiographic changes should always be interpreted in the clinical context of the individual patient.

It is important to acknowledge that several concurrent therapies administered during the stabilization period may have influenced both respiratory rate and echocardiographic measurements. All cats received supplemental oxygen, and most were mildly sedated with butorphanol prior to POCUS examinations. Oxygen therapy and sedation can independently reduce anxiety and respiratory rate and effort, contributing to the observed decrease in respiratory rate, independent of diuretic effects. Moreover, 11 of 25 cats underwent therapeutic thoracocentesis before the T3 timepoint, which likely resulted in immediate relief of dyspnea and a decrease in intrathoracic pressure. This intervention could transiently alter cardiac chamber preload, potentially contributing to the measured reduction in left atrial size. Therefore, although the significant changes in respiratory rate and left atrium dimensions after furosemide administration reflect the expected hemodynamic response to diuresis, the concurrent effects of oxygen supplementation, sedation, and thoracocentesis represent potential confounding factors that should be considered when interpreting these results. In addition to diuretic-induced preload reduction, other factors such as dehydration, or acute hemodynamic changes including intracardiac thrombus formation could contribute to reduced atrial size. These potential confounders further highlight the complexity of interpreting left atrial dimensions in acutely ill cats.

Normalization of respiratory rate was achieved in 50% of cats, while normalization of LAD was documented in 32% of the cohort. Interestingly, only a single cat achieved normalization of the LA:Ao. Previous research has shown that LA:Ao is associated with relatively high intra- and inter-observer variability, making it the least reproducible index of left atrial size; moreover, it is highly dependent on the operator’s experience [28]. This may have introduced measurement bias in our study. That is the reason that LAD was introduced as an additional echocardiographic parameter, as this measurement has been demonstrated to show lower intra- and inter-observer variability. Therefore, LAD is regarded as more reliable and reproducible variable to assess left atrial size than LA:Ao in both cats and dogs [28,29].

Furthermore, nFo significant correlation was found between normalization of respiratory rate and normalization of LAD, suggesting that clinical recovery does not necessarily parallel structural reversal of left atrial enlargement.

To our knowledge, this is the first prospective multicenter study to quantify short- term changes in left atrial size and respiratory rate following parenteral furosemide administration in cats with acute CHF. Our findings emphasize that echocardiographic indices obtained after stabilization may underestimate the true degree of cardiac enlargement present at presentation. This has direct implications for clinical decision-making, particularly in referred cats that have already received diuretics prior to echocardiographic evaluation.

This study has several limitations. First, the study’s timeframe was limited, with measurements taken only at baseline and three hours after furosemide administration. During this period, not all cats achieved full clinical stabilization, which may not fully capture the drug’s impact on respiratory distress and atrial reverse remodeling. Performing additional POCUS cineloops at multiple time points, especially when each cat reached clinical recovery, could have provided a more comprehensive understanding of the specific furosemide dose and the extent to which furosemide decreases left atrial size. Additionally, the variable furosemide dosing introduced variability, which may complicate analysis of the dose–response relationship. Unequal group sizes in the administered furosemide doses (3, 2, and 20 cats) rendered Kruskal–Wallis testing underpowered; therefore, dose was treated as a continuous variable in regression analysis. Another limitation concerns potential spectrum bias introduced by our inclusion criteria. Because cats with only mild left atrial enlargement (LA:Ao < 1.8 and LAD < 18 mm) were excluded, our findings mainly reflect the effects of furosemide in cats with moderate-to-severe left atrial dilation. Consequently, these results may not be fully generalizable to cats in the early or borderline stages of CHF, in which echocardiographic changes after diuretic therapy might be less pronounced. This selection bias could have led to an overestimation of the magnitude of left atrium size reduction observed in our study sample. Given the inherent variability in left atrial measurements among cats with CHF, the study may have been underpowered to detect small but potentially clinically relevant changes in left atrium size despite a priori sample size estimation. Another limitation is that intra- and inter-observer variabilities were not formally assessed. In addition, all offline measurements were performed by a single ECC resident to maintain consistency. However, this introduces potential interobserver bias, as repeatability between operators was not evaluated. Although efforts were made to minimize expectation bias, the reviewer was not fully blinded to examination timepoints (T0 vs. T3), as these were visible on the images provided from one of the participating centers in this multicenter study. Nonetheless, evaluations were performed according to predefined measurement criteria to reduce subjective interpretation. Future studies would benefit from involving multiple independent reviewers to better assess measurement reproducibility. The brief, two-hour supervised POCUS training with competency verification before study start helped standardize image acquisition, but variability in technique among clinicians may still have affected image quality. Additionally, because not every cat underwent a full echocardiographic examination performed by an EBVS^®^-recognized Cardiology Specialist, the underlying heart disease, as well as the presence of left atrial spontaneous echo-contrast or left auricular thrombus remained unknown.

Future studies incorporating larger cohorts, standardized dosing protocols, serial POCUS measurements may gain more information about challenges in diagnostics and therapy of feline CHF.

## 5. Conclusions

In this prospective multicenter study, administration of parenteral furosemide in cats with acute left-sided CHF was associated with significant short-term reductions and in some cases normalization of respiratory rate and left atrial size. Because concurrent interventions such as oxygen therapy, sedation, and thoracocentesis were also applied, these findings represent associations rather than direct causal effects of furosemide alone. Clinically, this underscores the importance of careful interpretation of echocardiographic findings in cats that have received diuretic therapy before referral, as post-treatment measurements may underestimate the true degree of left atrial enlargement. Consequently, clinicians should remain cautious when excluding CHF in referred cats that appear to have a normal or only mildly enlarged left atrium after diuretic administration.

## Figures and Tables

**Figure 1 animals-15-03267-f001:**
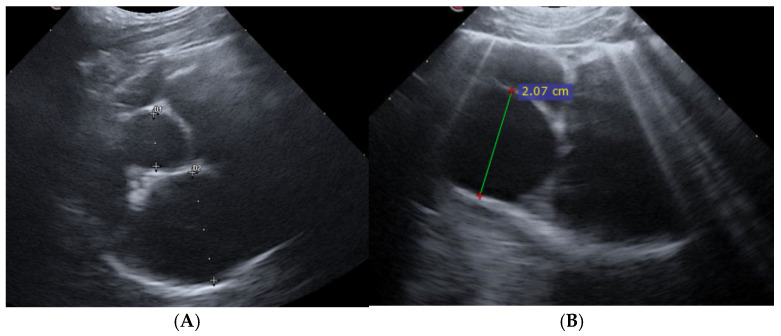
Two-dimensional Point-of- Care Ultrasound images showing the measurement of left atrium in right parasternal views at admission. (**A**) Measuring left atrium D2 and aorta D1 at a short axis view at maximum left atrial size to determine left atrial to aortic ratio (LA:Ao). In this example, the LA:Ao ratio measured 2.17. (**B**) Measurement of left atrial dimension (20.7 mm) on a right parasternal long axis 4-chamber image. Several pulmonary B-lines can also been appreciated.

**Figure 2 animals-15-03267-f002:**
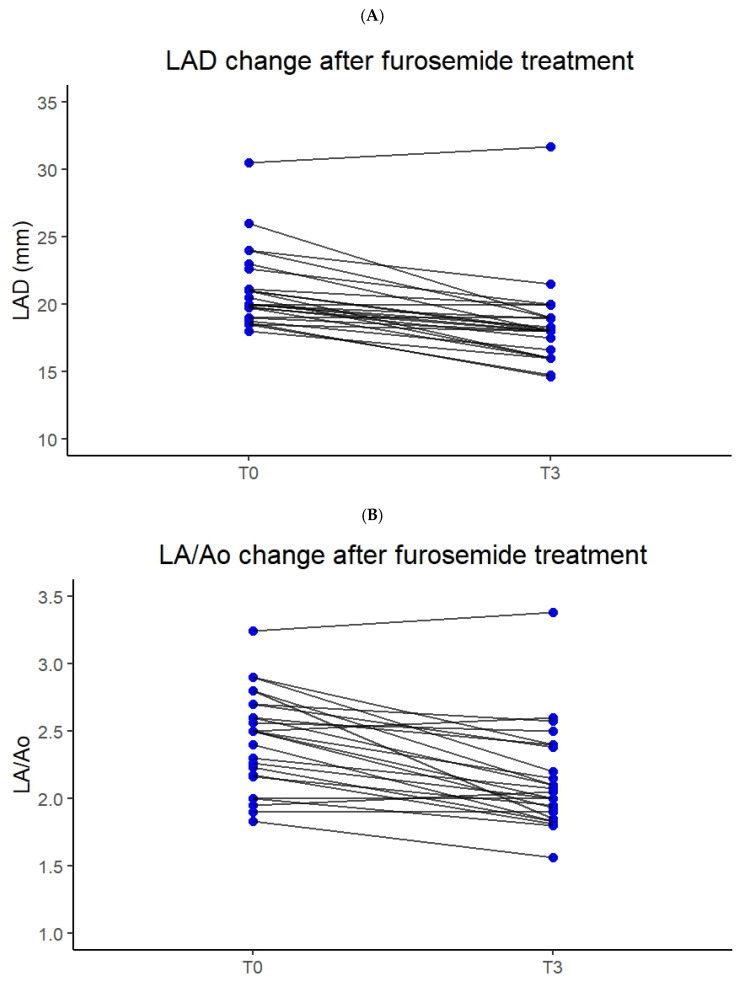
Spaghetti plots showing paired echocardiographic measurements for individual cats (n = 25) before (T0) and after (T3) parenteral furosemide administration. (**A**) Left atrial diameter (LAD). (**B**) Left atrium-to-aortic root ratio (LA:Ao). Each line connects values from the same cat at admission and after 3 h. The majority of cats showed a significant decrease in both LAD and LA:Ao (*p* < 0.001). One cat showed an increase in left atrium following treatment.

**Figure 3 animals-15-03267-f003:**
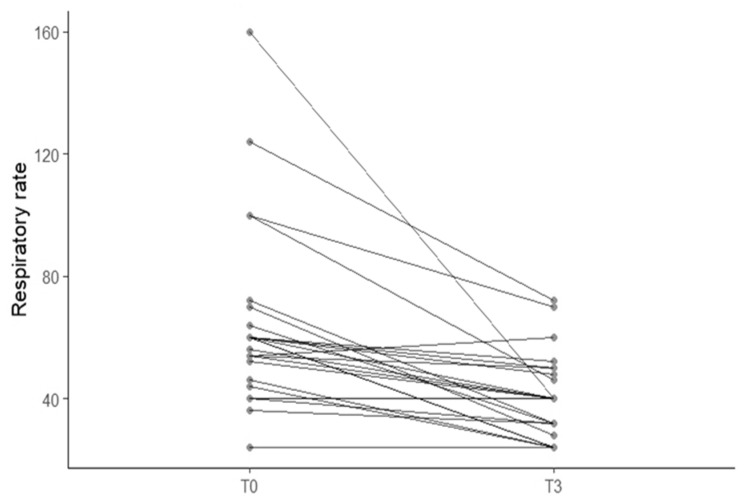
Spaghetti plot showing paired respiratory rate measurements for individual cats (n = 25) at baseline (T0) and at follow-up (T3). Each plot connects paired values from the same cat at admission and 3 h after intramuscular furosemide administration and oxygen therapy. The majority of cats showed a significant decrease in respiratory rate from T0 to T3 (*p* < 0.001).

## Data Availability

The data that support the findings of this study are available from the corresponding author upon reasonable request.

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
