# Peer review of "Acute Effect of Furosemide on Left Atrium Size in Cats with Acute Left-Sided Congestive Heart Failure"

_animals, 2025, doi:10.3390/ani15223267_

Round 1
Reviewer 1 Report
Comments and Suggestions for Authors
This is an interesting study with some clinically relevant information. However, some major concerns need to be addressed related to the interpretation of the data and confounding variables. See attached comments below.
- A) Major comments
- The results are inconsistently reported throughout the Abstract and Results; LAD is reported as “2.10 ± 28 mm” and “1.84 ± 32 mm”. Change to all cm or to mm.
- Cats received oxygen, sedation (butorphanol), thoracocentesis for 11/25, and serial furosemide; each can change RR and potentially LA loading conditions independent of diuresis. Did all cats receive thoracocentesis before T3? I would recommend adding discussion/clarification on the effect of concurrent therapies on RR and LA size.
- A comparison of ΔLAD across cumulative dose groups 3 mg/kg (n=3), 4 mg/kg (n=2), 5 mg/kg (n=20) using Kruskal–Wallis was reported. With such skewed groups, the test has minimal power and violates typical balance assumptions. Consider removing this inference and consider treating the furosemide dose as a continuous variable in a regression analysis.
- The study defined <40/min as a normal while cats are in oxygen and or received sedation. Please justify the threshold (citations typically use resting RR at home; clinic RR cutoffs may differ) and clarify that RR was measured in oxygen and after sedation, which complicates prognostic inference.
- Excluding cats with LA: Ao < 1.8 and LAD < 18 mm means only cats with clearly enlarged LAs at baseline were included. However, as the authors are aware, not every cat in CHF may meet the cutoff in borderline/early CHF. Please discuss spectrum bias and clinical implications for referred cats with mild LA enlargement.
- POCUS was performed by emergency veterinarians after a 2-hour training; measurements were reviewed by a single ECC resident. Please (i) report device models/probe frequency, frame rate if applicable; (ii) the repeatability of this measurement by the single ECC resident may have a major implication, and as such acknowledge this as a limitation, and clarify whether the reviewer was blinded to the timepoint to reduce expectation bias. Additionally, did the training involve any verification of performance, such as the ability to consistently obtain quality images or any competency measures?
- Lines 157 to 159: For the priori sample size estimation, the mean LA size used in calculation (any reference for this), and standard deviation, and the expected differences should be reported. Given the wide biological variability of LA size, I would have expected a much bigger sample size to document a reliable reduction in LA size. This should be clarified.
- Consider modifying the conclusion. The conclusion that furosemide “induces” LA size reduction is plausible but, given co-interventions, should be softened to “is associated with short-term reductions… while concurrent oxygen, sedation, and in many cases thoracocentesis may also contribute.” Emphasize the practical take-home: if a cat arrives post-diuretic, LA may underestimate disease severity, so weigh history and response to therapy in making the clinical diagnosis. You may highlight the percentage of cats that would have been misclassified as heart failure negative if LA: AO was assessed at T3 after furosemide injection.
- I would recommend removing the discussion about long-term survival, as the study was neither powered nor hypothesized to identify any relationship between furosemide dosing, change in LA size, with survival.
- B) Minor edits
Line 71 change “echocardioghraphy” to “echocardiography”.
Line 96 “bloodtest” → “blood test”; “NT-proBNP” → specify the assay brand/manufacturer
Line 77: “erroneously categorizing these cats of not having CHF” → “…as not having CHF.” (grammar).
Line 125: “normal or only mildly dilated left atrial size (LA:Ao < 1.8 and LAD < 18 mm)” → clarify if both of these criteria must be below threshold, or either one is sufficient. How was the discrepancy between the two cut-offs handled?
Line 113: There are several spellings, spacing, and grammatical errors throughout the manuscript. For eg. “thoracocenthesis” instead of “thoracocentesis”. It’s later spelled correctly. cautiosly” → “cautiously”; “veterinarin” → “veterinarian”; “recieved” → “received”. Please review all the spellings to ensure readability.
2.3 Follow-up: Define how deaths were ascertained; specify if phone follow-up or EMR only.
Author Response
Please see the attachment,
Sincerely
Sarah Miliaux

Reviewer 2 Report
Comments and Suggestions for Authors
Thank you for the opportunity to review the article “Acute effect of furosemide on left atrium size in cats with acute left-sided congestive heart failure.” The manuscript presents the volume response of the left atrium in recognition of measurements such as LAD and LA/Ao as basic parameters that may indicate heart failure. In addition, one of the important clinical parameters, such as respiratory rate, which is the basic unit of measurement used to assess the effectiveness of a given diuretic, was monitored. The article is well written, but I have many concerns about the paper.
General comments:
- The manuscript presents POCUS as a first-line procedure for assessing the chest and cardiac structures. It indicates that parameters such as LA/Ao and LAD are critical points in the assessment of CHF, suggesting that they can be significantly misleading, especially when the patient is admitted after initial stabilization with diuretics. While they are indeed extremely important, they cannot be defined as the two critical parameters in the assessment of CHF. Often, feline patients with an LA/Ao ratio equal to or greater than 2.5 or LAD 30 mm are not in CHF, and vice versa. They should be treated as one of the pieces (important pieces) of the puzzle that is the overall hemodynamic situation. In the vast majority of cases, due to the patient's condition, additional measurements are not possible, but a chest X-ray is a very helpful factor. I wonder why it was not included in the study. It provides a lot of information, is quick to perform and non-invasive, and also allows successful monitoring of the patient's condition after administration of furosemide - Ferasin, L., & Defrancesco, T. (2015). Management of acute heart failure in cats. Journal of Veterinary Cardiology, 17, S173-S189. In this manuscript, the authors nicely present different stages in the development of the lesion at the time of the radiographic study.
- I believe that, in addition to aggressive diuresis, which can lead to a rapid reduction in heart chambers, as indicated by the authors, other causes of this condition should also be indicated, despite the presence of CHF, such as hypovolaemia/dehydration due to poor clinical condition, and not necessarily increased diuresis caused by the administration of a diuretic, long-acting glucocorticoid injection in the preceding week or any acute exacerbating event such as a rapid formation of an intracavitary thrombus. Important to consider in the discussion and its extension.
- Left-sided CHF was diagnosed based on a history of sudden onset of increased respiratory rate or increased respiratory effort, as well as an abnormal point-of-care N-terminal pro-brain natriuretic peptide (NT-proBNP) blood test result – since this indicator was mentioned in the materials and methods section, why was it not discussed in the results and disscusision section?
- The examination and recording of cine loops was repeated three hours (T3) following parenteral furosemide + Their respiratory rate and effort were monitored hourly. Additional intravenous furosemide doses (1 mg/kg) were administered hourly (T1, T2, T3) as long as the respiratory rate remained above 40 breaths per minute or increased respiratory effort persisted in the oxygen cage, within the 3-hour study period. – Measurements were not taken until 3 hours after the first dose of furosemide. Why were echocardiographic parameters compared only 3 hours after furosemide administration? When administered intramuscularly, this drug is most effective 1 to 2 hours after administration. When administered intravenously, it is most effective after 30 minutes. Why was the decision made to take measurements after 3 hours?
Minor comments;
- Normalization of respiratory rate, defined as respiratory rate less than 40 breaths per minute at follow-up (T3), was achieved in 10 of 25 cats (50%). – 10 out of 25 cats is 40%. This error also occurs in the discussion
- Importantly, the finding that one third of cats in this cohort exhibited normalization of LAD within only three hours of parenteral furosemide administration indicates that left atrial size has to be evaluated cautiously in cats that received furosemide from the referring veterinarian before referral. As a result, CHF might be considered less likely cause of respiratory distress in these cats if imaging is performed after furosemide therapy. This may lead to diagnostic challenges, particularly in referred cats that received furosemide treatment prior to initial cardiac evaluation. – Therefore, it is important to perform a supporting examination in the form of X-rays. In addition to imaging various cotton fluff-like patterns such as nonuniformly diffuse, uniformly diffuse, multifocal, and focal. Furthermore, it may also indicate hydrothorax or ascites due to R-CFH. In addition, it is helpful in differential diagnosis of conditions such as lung disease, bronchial asthma, tumors, diaphragmatic-pericardial hernia, or diaphragmatic hernia, which may affect the respiratory pattern.
- Interestingly, one cat exhibited an increase in left atrium after furosemide treatment. Possible explanations include measurement variability or an inadequate therapeutic response within the short observation window. This emphasizes that echocardiographic changes should always be interpreted in the clinical context of the individual patient. "When one patient responded differently to the treatment, which was contrary to the overall results, I would ask for further discussion of what could have caused this condition in acute heart failure, e.g., decreased delivery to the kidney (R-CHF – kidney congestion, poor cardiac output), Decreased secretion into proximal convoluted tubule, enhanced Na absorption in distal nephron, etc.
Summary: The article presents results related to changes in left atrial size and RR during Acute CHF in response to administered doses of furosemide. These results are consistent with earlier reports on the effect of furosemide on heart chamber size in cats. In the vast majority of cases, this response did not lead to normalization of LAD (only 40% of patients) and only in 1 case was LA/Ao below 1.8. Furthermore, these results were not statistically significant in the context of clinical symptoms. Therefore, I would like to ask what information this study adds to the literature in the context of previous reports. The message related to the accurate evaluation of patients with suspected CHF is valid, but relying on individual parameters is not convincing in terms of monitoring and patient management, especially since the statistical results were not significantly related to the clinical picture of the subjects. The limitations section points to many weaknesses in the manuscript. In my opinion, correcting these limitations would help the manuscript gain much more value.
Author Response
Please see the attachment,
Sincerely,
Sarah Miliaux

Round 2
Reviewer 1 Report
Comments and Suggestions for Authors
I would like to thank the authors for making the requested changes. I only have a few pending suggestions.
Abstract lines 38-40: The decimal is still not placed correctly for the LA diameter. This has been changed in the body of the manuscript. Please correct
Lines 166-171: Survival information is removed from discussion. However, it is still presented in the results and in statistical methods. As previously pointed out, the study was not powered for survival analysis. Also, it is not intuitive to think that a change in RR or LA diameter in the first 3 hours of presentation will have a major impact on long-term survival. This is shown in your short-term survival results, i.e, there was no impact. But this could also be a lack of power issue. In veterinary medicine, unfortunately, similar reporting survival is prevalent. Consider completely removing all information related to survival throughout the manuscript. This will make your manuscript clean and succinct.
Lines 197-199: Clarify which regression model was used. Did we consider using individual animals as a random effect?
Author Response
Please see the attachment,
Kind regards

Reviewer 2 Report
Comments and Suggestions for Authors
I would like to kindly thank the authors for answering my questions. In the future, defining an X-ray pattern for signs of heart failure and monitoring the response to a given therapy along with echocardiographic changes would significantly enrich the content. Furthermore, considering the kinetics and hemodynamics of furosemide, shorter intervals for validation of atrial size should be taken into account, as the response to intramuscular injection of the dose is faster than after 3 hours, especially with repeated doses. Apart from that, I have no further comments. Thank you for the opportunity to review this article.
Author Response
Please see the attachment,
Kind regards
